# Novel Biphenyl Amines Inhibit Oestrogen Receptor (ER)-α in ER-Positive Mammary Carcinoma Cells

**DOI:** 10.3390/molecules26040783

**Published:** 2021-02-03

**Authors:** Basappa Basappa, Baburajeev Chumadathil Pookunoth, Mamatha Shinduvalli Kempasiddegowda, Rangappa Knchugarakoppal Subbegowda, Peter E. Lobie, Vijay Pandey

**Affiliations:** 1Laboratory of Chemical Biology, Department of Studies in Organic Chemistry, University of Mysore, Manasagangotri, Mysore 570006, India; salundibasappa@gmail.com; 2Department of Chemistry, Bangalore University, Bangalore 560001, India; baburajeevnambiar@gmail.com; 3Tsinghua Berkeley Shenzhen Institute, Tsinghua Shenzhen International Graduate School, Tsinghua University, Beijing 518055, China; mamathagowdask@gmail.com; 4Institution of Excellence, University of Mysore, Manasagangotri, Mysore 570006, India; rangappaks@gmail.com; 5Institute of Biopharmaceutical and Health Engineering, Tsinghua Shenzhen International Graduate School, Tsinghua University, Beijing 518055, China; 6Shenzhen Bay Laboratory, Shenzhen 518055, China

**Keywords:** arylamination, ERα ligands, human breast cancer cells, oestradiol, adamantane

## Abstract

Herein, the activity of adamantanyl-tethered-biphenyl amines (ATBAs) as oestrogen receptor alpha (ERα) modulating ligands is reported. Using an ERα competitor assay it was demonstrated that ATBA compound 3-(adamantan-1-yl)-4-methoxy-N-(4-(trifluoromethyl) phenyl) aniline (AMTA) exhibited an inhibitory concentration 50% (IC_50_) value of 62.84 nM and demonstrated better binding affinity compared to tamoxifen (IC_50 =_ 79.48 nM). Treatment of ERα positive (ER+) mammary carcinoma (MC) cells (Michigan Cancer Foundation-7 (MCF7)) with AMTA significantly decreased cell viability at an IC_50_ value of 6.4 μM. AMTA treatment of MC cell-generated three-dimensional (3D) spheroids resulted in significantly decreased cell viability. AMTA demonstrated a superior inhibitory effect compared to tamoxifen-treated MC cell spheroids. Subsequently, by use of an oestrogen response element (ERE) luciferase reporter construct, it was demonstrated that AMTA treatment significantly deceased ERE transcriptional activity in MC cells. Concordantly, AMTA treatment of MC cells also significantly decreased protein levels of oestrogen-regulated CCND1 in a dose-dependent manner. In silico molecular docking analysis suggested that AMTA compounds interact with the ligand-binding domain of ERα compared to the co-crystal ligand, 5-(4-hydroxyphenoxy)-6-(3-hydroxyphenyl)-7- methylnaphthalen-2-ol. Therefore, an analogue of AMTA may provide a structural basis to develop a newer class of ERα partial agonists.

## 1. Introduction

17β-oestradiol (E2), a steroid hormone, plays important roles in the regulation of growth, differentiation, and function of a wide array of target tissues in both the male and female reproductive tracts, mammary glands, and skeletal and cardiovascular systems [1,2]. E2 exerts biological effects through at least two types of oestrogen receptors (ERs), namely ER-alpha (α) and ER-beta (β) [3,4]. These two receptors exhibit a high degree of homology in their ligand-binding and DNA-binding domains [5,6,7]. However, there are considerable differences in the N-terminus region of the two receptors [8]. The exact roles of ERβ are not clear, but it is known to counteract the activities of ERα [7].

Elevated levels of E2 are significantly associated with the neoplastic transformation and progression of female reproductive-related malignancies such as endometrial [9,10], ovarian [10], and breast cancer (BC) [11,12]. About two-thirds of human BC cases are ERα positive (+) [13]. Upon binding to E2, ERα dissociates from molecular chaperone complexes, dimerizes, migrates to the nucleus, and binds to specific DNA sequences (oestrogen response element (ERE)) that regulates the transcription of genes vital for BC cell survival [14] and metastasis [8,15]. In addition, E2 along with TGFβ1 enrich cancer stem cell populations in BC, leading to increased migration and resistance to therapy [16].

ERα is therefore an important therapeutic target in breast cancer with drugs limiting estrogenic activity to delay cancer progression [17,18,19]. Three approaches have been utilized clinically to inhibit ERα-related function. First, selective oestrogen modulators (SERMs) (e.g., tamoxifen, raloxifene) have been used to competitively bind to ER and displace E2, consequently inhibiting downstream signalling [17]. Second, selective ER degraders (SERDs) (e.g., fulvestrant) have been used to selectively promote ER degradation [18,19]. Finally, aromatase inhibitors have been used to inhibit aromatase enzyme activity and subsequently decrease the aromatization of androgens into oestrogens [20].

The majority of ERα-based anticancer agents such as tamoxifen, toremifene, raloxifene, and fulvestrant have been observed to be well tolerated in the clinic [21]. However, prolonged use of tamoxifen and toremifene in MC is associated with increased risk of endometrial cancer and eventual resistance to therapy leading to relapse [22,23]. Raloxifene use was also observed to be associated with increased risk of deep vein thrombosis in MC patients [24]. Therefore, to provide potential therapeutic alternatives, the discovery of novel ERα-targeting compounds is still required.

Synthetic oestrogens such as 4-(1-adamantyl) phenol (AdP) and 4,4-(1,3-adamantanediyl) diphenol (AdDP) bind to ERs and stimulate ER activity. Using structural references from synthetic oestrogens such as 4-(1-adamantyl) phenol (AdP) and 4,4-(1,3-adamantanediyl) diphenol (AdDP), a novel library of analogous AdDPs called adamantanyl-tethered-biphenyl amines (ATBAs) that potentially target ERα was designed [25]. We herein explored the in vitro and ex vivo inhibitory activity of newly designed ATBA compounds that bear phenolic amine as a linkage (unlike tamoxifen, GW368, and AdDP, small molecules which contain phenolic/ether linkages) as a new ligand that targets ERα in MC cells (Figure 1).

## 2. Results and Discussion

Diverse functionalized ATBA compounds were synthesized by treating adamantine bromo compound (**1a**) with substituted amines via an arylamination reaction (Scheme 1) [26]. The chemical synthesis and characterization of these compounds were reported recently [26].

First, to rank the novel biphenyl derivatives based on binding affinity to ERα, a commercially available fluorescence polarization assay kit (PolarScreen™ ER Alpha Competitor assay) was utilized. ATBA compounds were analysed for their capacity to displace fluorescently labelled E2 to form an ER-Fluormone ES2 complex. The smaller size of the free molecule can be discerned by increased mobility in the solution, which is detected by fluorescence polarization (FP). A shift in the FP value in the presence of the test compounds is used to determine the relative affinity of test compounds for ERα. The competitive binding affinity of the novel biphenyl derivatives is summarized in Table 1. Based on the FP value, the binding affinity of compound **4c** was observed to be 62.84 nM, whereas for tamoxifen it was observed to be 79.48 nM. E2 exhibited a binding affinity of 6.27 nM. ATBA compound **4c** exhibited higher binding affinity when compared to tamoxifen, but it was lower than that of E2. Therefore, the addition of –CF3, -OH, or -CH3 group tethered amines improved the binding affinity of the ATBA compounds to ERα.

Next, we determined the potential functional effects of novel ATBA compounds on the viability of MCF7 cells using an AlamarBlue^®^ cell viability assay. The different ATBAs reduced the viability of MCF7 cells to differing degrees (Table 1). Paralleling the outcomes of the cell viability assay with ER-α binding, compounds **4a** and **4b** slightly reduced cell viability in MC cells with an IC_50_ greater than 50 µM. However, when the chlorine present in compound **4a** or the methoxy group present in compound **4b** was replaced with the -CF_3_ group in compound **4c**, the loss of cell viability increased dramatically with an IC_50_ value of 6.29 µM. The presence of the hydroxyl group in compound **4d** and methyl group in compound **4f** also showed a better reduction of cell viability with IC_50_ at 8.81 µM and 12.3 µM, respectively. Similarly, the replacement of the benzene ring with a pyridine nucleus in compound **4g** also showed activity with an IC_50_ of 12.09 µM. Other compounds **4h**, **4i**, **4k**, and **4l** reduced cell viability but with IC_50_ values close to 50 µM or higher. We also determined the functional efficacy of the **4c** compound in additional ER+ MC cell lines using an AlamarBlue^®^ cell viability assay. Compound **4c** decreased cell viability of T47D cells with an IC_50_ value of 7.38 + 1.79 μM; BT474 cells with an IC_50_ value of 6.36 + 2.07 μM; and MDA-MB-361 cells with an IC_50_ value of 8.31 + 3.52μM.

Furthermore, we examined the effect of the novel biphenyl derivatives on preformed spheroids generated using MC cells in 3D Matrigel (Figure 2). Of the biphenyl derivatives tested, 4c-, 4d-, 4f-, and 4g-treated MC cell-generated spheroids exhibited significantly decreased cell viability compared to vehicle (DMSO) treated MC cell-generated spheroids. Among biphenyl derivatives, only the 4c derivative demonstrated a significant inhibitory effect compared to TAM-treated MC cell-generated spheroids.

Next, to evaluate the activity of the synthesized compounds as agonists or antagonists to ERα, a reporter assay was performed with ERE-luc [22,27]. The results of ERE activity (inhibition (upper chart) and activation (lower chart)) in response to the compounds are summarized in Figure 3. Of the biphenyl derivatives tested, 4c, 4d, 4f, and 4g exhibited the most potent antagonist activity against ERα in a dosage-dependent manner, represented by the most significantly decreased luciferase activity. Tamoxifen was used as positive control. The IC_50_ value of 4c was 0.83 ± 0.06 µM compared to TAM 2.07 ± 0.09 µM, as measured using ERE reporter assay. None of the tested biphenyl derivatives demonstrated a significant agonistic effect.

Increased expression of CCND1 is associated with approximately 50% of breast cancer cases and directly regulated by ER signalling [27,28]. To determine functional effects downstream of ERα, we examined the effect of ATBA compounds on *h*CCND1 protein expression in MC cells. Amongst the tested ATBA compounds, compound **4c** most potently decreased the protein levels of CCND1 in MC cells in the nanomolar range when compared to other ATBA compounds (Figure 4).

Molecular docking analysis was performed by using the co-crystal structure of the hERα ligand-binding domain (LBD) with the naphthalene based small molecule, GW368 (5-(4-hydroxyphenoxy)-6-(3-hydroxyphenyl)-7-methylnaphthalen-2-ol; PDB ID: 3DT3) [31]. CDOCKER program of Accelrys DS version 2.5 was utilized. The receptor was energy-minimized and used for molecular docking studies as described previously [32,33]. The docking analysis observed that the most active compound **4c** bound to the LBD of ERα with a CDOCKER interaction energy of 31kcal/mole and its binding mode was almost similar to GW368 with a perturbed trifluoromethyl (-CF3) group of compound 4c (Figure 5). The bulky adamantyl moiety of compound **4c** may fill spaces in the LBD that are unoccupied when GW368-like molecules are bound. These relatively novel structural features of ATBAs could account for their high affinity for ERα. Thus, the addition of a -CF3 group to adamantyl-tethered-amino biphenylic, derivative 4c, shows improved efficacy and stability for further development as a potential therapeutic for ER + MC.

To determine the potential in vivo utility of the ATBA compounds, 15 properties associated with ADMET were determined by using the vNN-ADMET platform for the most active compounds (4c, 4d, 4g, 4h, and TAM as a comparison) [34]. The responses from the vNN-ADMET platform are tabulated in Table 2. The in silico analyses of ATBA compounds predicted that all of the active compounds will not be hepatotoxic, will not exhibit cytotoxicity, will not be metabolized rapidly by human liver membranes, will not inhibit drug metabolizing CYP450s, may pass through the blood-brain barrier (BBB), will not be a P-glycoptrotein inhibitor but may be a substrate, may exhibit hERG activity (as for TAM), will not impact matrix metalloproteinases, and will not exhibit chemical mutagenicity (Table 2). In addition, the predicted maximum recommended therapeutic dose of compound 4c was predicted to be approximately four times higher as compared to TAM.

## 3. Materials and Methods

The human Michigan Cancer Foundation-7 (MCF7) cell line was obtained from the American Type Culture Collection (ATCC, Rockville, Gaithersburg, MD, USA) and was cultured as per ATCC propagation instructions.

ER-alpha competitive binding assay: A competitive binding assay was performed using 17β-oestradiol and the PolarScreen™ ER Αlpha Competitor assay kit (Life Technologies^TM^, Carlsbad, CA, USA). The assay was performed as per the manufacturer’s instructions. The Fluorescence polarization value (mP) of each well on a fluorescence polarization plate was measured using fluorescence polarization Tecan infinite M1000PRO multimode microplate reader (Tecan, Switzerland). Data were modelled using GraphPad Prism^®^ software from GraphPad Software, Inc. [27,30].

AlamarBlue^®^, 3D Matrigel, and luciferase assay: AlamarBlue^®^ cell viability kits were obtained from ThemoFisher Scientific (Waltham, Massachusetts, USA), and the assay was performed as previously described [22,35]. 3D Matrigel assays were performed as previously described [30]. Luciferase assays were performed as previously described [22,35]. Briefly, transfections were carried out in triplicate using 1 μg of the appropriate luciferase reporter construct or empty vector along with 0.1 μg of Renilla luciferase construct as a control for transfection efficiency. Luciferase activities were measured using the Dual-Luciferase Assay System (Promega Corp, Madison, WI, USA).

Western blot analysis: Western blot analysis was performed as previously described [22,35], using a primary monoclonal antibody against CCND1 and β-ACTIN obtained from Cell Signalling, Danvers, Massachusetts, USA. Quantification of blot was performed by use of ImageJ software from NIH, USA (http://imagej.nih.gov/ij/) as described previously [27,29,30].

In silico studies: Molecular in silico interaction was carried out using 2DT3 protein ID and the ATBA compounds as ligands. The accelrys DS 2.5 was used for this study. After molecular redocking of *GW368*, we docked ATBAs to the ligand binding domain of ER using the CDOCKER programme as we reported in our previous publications [32]. Results were analysed using the accelrys visualization platform. The publicly available vNN Web Server was used for the ADMET predictions for the most active compounds and tabulated (https://vnnadmet.bhsai.org/).

## 4. Conclusions

It is herein disclosed that the ATBA compound **4c** exhibits binding to ERα with promising inhibition of oestrogenic functions in *h*ERα positive MC cells. In silico molecular docking studies revealed that compound **4c** bound to the ligand-binding domain (LBD) of ERα strongly when compared to the co-crystal ligands. Further investigations of compound 4c are warranted to determine its pharmacological features and potential in vivo utility.

## Data Availability

The data presented in this study are available on request from the corresponding author.

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
