# Peer review of "Novel Biphenyl Amines Inhibit Oestrogen Receptor (ER)-α in ER-Positive Mammary Carcinoma Cells"

_molecules, 2021, doi:10.3390/molecules26040783_

Round 1

Reviewer 1 Report

In the proposed work, the authors describe and characterize a new class of molecules capable of binding the ERa receptor and test the ability of these molecules to induce cell death in MCF7 breast cancer cells.

The major limitation of this work remains the fact that among all those proposed only one molecule seems to have a similar or slightly greater efficiency than the well-known Tamoxifen at least in MCF7.

Major points

The Authors should first complete Table 1 by adding the Cell viability IC50 for tamoxifen with their experimental setting since the other values are very similar even for the most effective molecule (4c)

To give a greater value to the results, the authors should confirm the efficiency of at least the 4c ​​molecule on another cell line similar to MCF7.

Moreover, to understand if the proposed molecular backbone is really developable for an in vivo application, the authors should carry out at least one stability test with microsomes.

Minor points

-Please check lane 27 of the abstract.

-Please check carefully the english example lane 57 Three approaches have been utilized to inhibit activity clinically.

- Please change lane 60 explaining the function of aromatase

-Please change Figure 4; considering the importance of the figure the authors should mediate the values of 3 independent experiments in the right panel

Author Response

Response to reviewer

Reviewer 1:

In the proposed work, the authors describe and characterize a new class of molecules capable of binding the ERa receptor and test the ability of these molecules to induce cell death in MCF7 breast cancer cells. The major limitation of this work remains the fact that among all those proposed only one molecule seems to have a similar or slightly greater efficiency than the well-known Tamoxifen at least in MCF7.

Response: We thank the reviewer for the constructive, and positive overview of the manuscript.

Major points

  1. The Authors should first complete Table 1 by adding the Cell viability IC50 for tamoxifen with their experimental setting since the other values are very similar even for the most effective molecule (4c)

Response: As per the reviewer’s request the IC50 value (6.92 +2.13) of tamoxifen in now represented in Table 1.

  1. To give a greater value to the results, the authors should confirm the efficiency of at least the 4c ​​molecule on another cell line similar to MCF7.

Response: In concordance with the reviewer, we have now incorporated new information in the revised manuscript, as below;

“We also determined the functional efficacy of the 4c compound in additional ER+ MC cell lines using an AlamarBlue® cell viability assay. Compound 4c decreased cell viability of T47D cells with an IC50 value of 7.38 +1.79mM; BT474 cells with an IC50 value of 6.36 +2.07mM; and MDA-MB-361 cells with an IC50 value of 8.31 +3.52mM.”

  1. Moreover, to understand if the proposed molecular backbone is really developable for an in vivo application, the authors should carry out at least one stability test with microsomes.

Response: This is a constructive suggestion and we are agreed with the reviewer. However, this suggested experiment is out of the scope of the current study. The main aim of this study is to identify the functional core structure, which will provide the basis for the further development of novel library targeting ER signalling associated effectors. Therefore, a detailed pharmacological based investigation such as pharmacokinetics, pharmacodynamics, toxicity, dose formulations, route of administration, solubility, microsomal stability, maximum tolerance dose, etc., will be a major goal for the screened molecules of the newer library, which will be generated based on the core structure of compound 4c.  We have however indicated the potential utility of the molecular backbone with use of in silico ADMET predictions in comparison to TAM (see Table 3)

Minor points

  1. -Please check lane 27 of the abstract.

Response: In concordance with the reviewer, we have now revised the information represented in line 27 of the abstract in the revised-manuscript, as below;

“Subsequently, by use of an estrogen response element (ERE) luciferase reporter construct, it was demonstrated that AMTA treatment significantly deceased ERE transcriptional activity in MC cells.”

  1. -Please check carefully the english example lane 57 Three approaches have been utilized to inhibit activity clinically.

Response: We acknowledge the reviewer’s concern. Therefore, we have modified the textual information in lane 57 in the revised-manuscript, as below;

“ERα is therefore an important therapeutic target in breast cancer with drugs limiting estrogenic activity to delay cancer progression [17-19]. Three approaches have been utilized clinically to inhibit ERα-related function . Firstly, the selective estrogen modulators (SERMs) (eg Tamoxifen, Raloxifene), which competitively bind to ER and displaces E2 and consequently inhibit downstream signalling [20]. Secondly, selective ER degraders (SERDs) (eg Fulvestrant), which selectively promotes ER degradation [19, 21]. And finally, aromatase inhibitors, which inhibit aromatase enzyme activity and subsequently decreases the aromatization of androgens into estrogens [22]..”

  1. Please change lane 60 explaining the function of aromatase

Response: We acknowledge the reviewer’s concern. Therefore, we have modified the textual information in lane 60  in the revised-manuscript, as above response to point 5.

  1. -Please change Figure 4; considering the importance of the figure the authors should mediate the values of 3 independent experiments in the right panel

Response: As per the reviewer’s suggestion we have modified the right panel of Figure 4 and represented the average of three independent analyses of CCND1 and β-ACTIN as in shown below (right side);

Reviewer 2 Report

First, it is difficult to evaluate the true significance of this report due to all the formatting issues and other problems in this submission. See below:

1.Abstract 2nd sentence missing characters...ER ____ competitor...

and other missing symbols in the abstract and MS

also please take out "superior" and replace with "significant" here in this the conclusions

2.The problem with this submission is it starts on page 2 in mid paragraph, please start with the Introduction

3.Figure 1 legend please correct and use full name for AdDP & ATBAs, etc.

4.The Introduction (near the end) needs a clearly stated hypothesis

5. Figure 3 is difficult to follow graphically (increase the width of the bars so the reader can see what is being displayed).

6. While 4c appears to have great efficacy compared to TAM, what other studies needs to follow in order to show the safety and effectiveness in therapeutic human applications, since TAM has such a long track record as a therapeutic drug?

7. Please discussion the comparisons between TAM and 4c for dosing, route of administration, potential side effects, and tolerance, etc.

Author Response

Reviewer 2:

First, it is difficult to evaluate the true significance of this report due to all the formatting issues and other problems in this submission. See below:

  1. Abstract 2nd sentence missing characters...ER ____ competitor...

and other missing symbols in the abstract and MS

also please take out "superior" and replace with "significant" here in this the conclusions

Response: First, we thank the reviewer for the constructive, and positive overview of the manuscript. We assure the reviewer that all misrepresentation of symbols in the abstract is due to the journal’s formatting platform. However, we have incorporated all the changes suggested in the revised manuscript. Also, the term “superior” is now excluded from the conclusion section in the revised manuscript.

  1. The problem with this submission is it starts on page 2 in mid paragraph, please start with the Introduction

Response: In our initial submission, the abstract and introduction sections were represented on separate pages. Again, this is an issue raised by the journal’s formatting platform.

  1. Figure 1 legend please correct and use full name for AdDP & ATBAs, etc.

Response: As per the reviewer’s suggestion, abbreviation of the compounds has been replaced with the International Union of Pure and Applied Chemistry (IUPAC) name, as below;

“Figure 1: Structures of the ER modulators; (A) 4,4-(1,3-adamantanediyl) diphenol; (B) and (C) newly synthesized adamantanyl-tethered-biphenyl amines that are used as ER ligands.”

  1. The Introduction (near the end) needs a clearly stated hypothesis

Response: As per the reviewer’s suggestion, the statement defining the hypothesis is now incorporated in the revised manuscript, as below;

“Therefore, to provide potential therapeutic alternatives the discovery of novel ERα-targeting compounds is still required”

  1. Figure 3 is difficult to follow graphically (increase the width of the bars so the reader can see what is being displayed).

Response: As per the reviewer’s request, we have modified the representation of Figure 3, as below;

Figure 3. Inhibition of Estradiol (E2) stimulated ERE transcriptional activity by the ABTA compounds (upper panel) and effect of ATBA compounds on ERE transcriptional activity in the absence of E2 (lower panel). MCF7 cells were co-transfected with ERE-luc (firefly luciferase) and Renilla luciferase construct as described in materials and methods. Cells were incubated with 17β Estradiol (E2) (10-8M) (left panel) and ATBA compounds (10-10 to 10-4M) and without Estradiol (lower panel). Results are shown as mean (+SD for triplicate transfection). TAM, Tamoxifen.

  1. While 4c appears to have great efficacy compared to TAM, what other studies needs to follow in order to show the safety and effectiveness in therapeutic human applications, since TAM has such a long track record as a therapeutic drug?

Response: As per reviewer 1 point 3 above.

  1. Please discussion the comparisons between TAM and 4c for dosing, route of administration, potential side effects, and tolerance, etc.

Response: As per reviewer 1 point 3 above.

Round 2

Reviewer 1 Report

all the comments have been addressed

Author Response

Not applicable

Reviewer 2 Report

The revised version of the MS was reviewed:

1.The introduction still does not contain a true “hypothesis.” Lines 64-65 and 70-71

Rather than providing a descriptive “validated” statement it should be a question. 

For example, and this is only an example- The purpose of this investigation was to examine whether ABTA compounds via Erα mediated actions in mammary cells (MC) displayed inhibitory activities using in vitro and ex vivo methods.

  1. Figure 1, still does NOT describe clearly what is being displayed please provide the names for each compound.
  2. Line 78 reported recently twice, please correct
  3. Scheme 1 does not clearly describe what this compounds are…..
  4. Table 1, please provide the chemical names for each of the compounds listed: 4a thru 4i
  5. Figure 3, top is still difficult to read, please increase the width of each bar in the histograms (see Figure 4).

Author Response

Response to reviewer

Reviewer 2:

  1. The introduction still does not contain a true “hypothesis.” Lines 64-65 and 70-71Rather than providing a descriptive “validated” statement it should be a question. For example, and this is only an example- The purpose of this investigation was to examine whether ABTA compounds via Erα mediated actions in mammary cells (MC) displayed inhibitory activities using in vitro and ex vivo methods.

Our Response: We thank the reviewer for his valuable comments. We wish to bring it to the reviewer notice that, adamanatane based diphenolic ligands were reported as ER alpha synthetic ligands. We revised the manuscript as “Unlike Tamoxifen, GW368, or AdDP, small molecules containing phenolic ether linkages, we herein explored the in vitro and ex vivo inhibitory activity of newly designed ABTA compounds, which bearing phenolic amine as a linkage and a new ligands that targets ERα in MC cells (Figure 1)”.

  1. Figure 1, still does NOT describe clearly what is being displayed please provide the names for each compound.

Response: As per the reviewer’s suggestion, we incorporated the name of each compound represented in the figure 1.

  1. Line 78 reported recently twice, please correct

Response: As per the reviewer’s suggestion, the error rectified in the revised manuscript.

  1. Scheme 1 does not clearly describe what this compounds are…..

Response: We thank the reviewer. We provided the descriptions accordingly.

  1. Table 1, please provide the chemical names for each of the compounds listed: 4a thru 4i

Response: Revised the manuscript accordingly.

  1. Figure 3, top is still difficult to read, please increase the width of each bar in the histograms (see Figure 4).

Response: As per the reviewer’s suggestion, we modified the figure 3 for the clear representation, as below;
